# The Local Activation of Toll-like Receptor 7 (TLR7) Modulates Colonic Epithelial Barrier Function in Rats

**DOI:** 10.3390/ijms24021254

**Published:** 2023-01-09

**Authors:** Javier Estévez, Vicente Martínez

**Affiliations:** 1Department of Cell Biology, Physiology and Immunology, Universitat Autònoma de Barcelona, 08193 Barcelona, Spain; 2Neuroscience Institute, Universitat Autònoma de Barcelona, 08193 Barcelona, Spain; 3Centro de Investigación Biomédica en Red de Enfermedades Hepáticas y Digestivas (CIBERehd), Instituto de Salud Carlos III, 28049 Madrid, Spain

**Keywords:** epithelial barrier function, host–bacterial interactions, imiquimod, intestinal permeability, tight junctions, TLR, Toll-like receptors

## Abstract

Toll-like receptors (TLRs)-mediated host–bacterial interactions participate in the microbial regulation of gastrointestinal functions, including the epithelial barrier function (EBF). We evaluated the effects of TLR7 stimulation on the colonic EBF in rats. TLR7 was stimulated with the selective agonist imiquimod (100/300 µg/rat, intracolonic), with or without the intracolonic administration of dimethyl sulfoxide (DMSO). Colonic EBF was assessed in vitro (electrophysiology and permeability to macromolecules, Ussing chamber) and in vivo (passage of macromolecules to blood and urine). Changes in the expression (RT-qPCR) and distribution (immunohistochemistry) of tight junction-related proteins were determined. Expression of proglucagon, precursor of the barrier-enhancer factor glucagon-like peptide 2 (GLP-2) was also assessed (RT-qPCR). Intracolonic imiquimod enhanced the EBF in vitro, reducing the epithelial conductance and the passage of macromolecules, thus indicating a pro-barrier effect of TLR7. However, the combination of TLR7 stimulation and DMSO had a detrimental effect on the EBF, which manifested as an increased passage of macromolecules. DMSO alone had no effect. The modulation of the EBF (imiquimod alone or with DMSO) was not associated with changes in gene expression or the epithelial distribution of the main tight junction-related proteins (occludin, tricellulin, claudin-2, claudin-3, junctional adhesion molecule 1 and Zonula occludens-1). No changes in the proglucagon expression were observed. These results show that TLR7 stimulation leads to the modulation of the colonic EBF, having beneficial or detrimental effects depending upon the state of the epithelium. The underlying mechanisms remain elusive, but seem independent of the modulation of the main tight junction-related proteins or the barrier-enhancer factor GLP-2.

## 1. Introduction

The disruption of the epithelial barrier function (EBF) with increased intestinal permeability is a common finding in several gastrointestinal disorders, including inflammatory bowel disease (IBD) and irritable bowel syndrome (IBS) [1,2,3,4]. The intestinal epithelium represents a functional barrier between the lumen and the internal milieu, allowing selective exchange processes (secretory and absorptive) and acting at the same time as a protective barrier against luminal antigens [5]. Gut commensal microbiota, considered for long time to have been a passive component of the gastrointestinal tract, is now considered as a dynamic player in intestinal homeostasis [6,7,8,9]. Gut microbiota and microbial-derived products act as luminal factors that interact continuously with the host. In physiological conditions, these interactions contribute to the maintenance of intestinal homeostasis. However, in states of dysbiosis, host–microbial interactions change and might lead to the activation of the immune system and the generation of a persistent inflammatory state with functional changes, including alterations in the EBF [10]. In this sense, dysbiosis and altered epithelial permeability are co-existent features of IBS and IBD [1,3,11,12,13,14,15]. 

Host–microbial interactions are largely mediated through pattern recognition receptors (PRRs). PRRs are expressed within the gut and recognize conserved microbial components named pathogen-associated molecular patterns (PAMPs) [16]. In particular, recognition of gut microbiota depends largely on the interaction with a subgroup of PRRs named Toll-like receptors (TLRs), with up to 13 members described in mammals (TLR1–13) [17,18]. The relative expression of the different TLRs differs depending upon the cell type and the cellular localization [18,19]. Within the gut, some of the predominant TLRs are TLR2, TLR3, TLR4, TLR5 and TLR7, being present in several cell types, including the epithelial cells [17]. TLRs directly interact with luminal microbial components and, upon activation, elicit neuroimmune responses that affect intestinal functions, including the EBF [20,21,22,23,24,25]. For instance, we have shown that during antibiotic-induced dysbiosis, host–bacterial interactions are altered, including changes in the expression of TLRs, and there are local changes in immune- and sensory-related systems within the gut [26,27]. Similar alterations might be elicited when TLRs are over-stimulated, mimicking a state of dysbiosis with altered host–bacterial interactions. In this sense, we have shown that direct stimulation of colonic TLR7 with the selective agonist imiquimod leads to immune- and sensory-related changes similar to those observed during intestinal dysbiosis [28,29]. Moreover, imiquimod reduced the serum levels of inflammatory cytokines and ameliorated clinical signs and intestinal lesions during DSS-induced colitis in mice [30]. 

Taking these observations into account, the aim of the present work was to directly assess the effects of colonic TLR7 stimulation on the EBF in rats. For this, we assessed the effects of the over-stimulation of colonic TLR7 with the selective agonist imiquimod on epithelial electrical parameters and on epithelial permeability to macromolecules in in vitro (the Ussing chamber system) and in vivo conditions. In order to characterize the underlying mechanisms mediating potential TLR7-dependent modulation of the EBF, we also assessed changes in the expression and epithelial distribution of the main tight-junction (TJ)-related proteins (occludin, tricellulin, claudin-2, claudin-3, junctional adhesion molecule 1 -JAM-1- and zonula occludens-1 -ZO-1-). Moreover, changes in the expression of the barrier-enhancer factor glucagon-like peptide 2 (GLP-2, assessed through the expression of its precursor, proglucagon) [31] were also assessed. Finally, the induction of a colonic inflammatory-like state associated with the over-stimulation of TLR7 was assessed at the macroscopical and molecular levels.

## 2. Results

### 2.1. The Acute Addition of Imiquimod to the Ussing Chamber Did Not Affect Colonic Electrical Parameters or Permeability to Macromolecules

After stabilization, the short circuit current (Isc), conductance (G) and potential difference (PD) had a value of −26.1 ± 2.4 µA cm^−2^, 23.3 ± 1.7 ms cm^−2^ and −1.2 ± 0.1 mV, respectively (*n* = 24 tissue samples from 5–7 animals). These values are similar to those previously described by us in equivalent experimental conditions [12,32]. Neither the apical nor the basolateral addition of imiquimod (125 µM) to the Ussing chamber affected the basal electrical parameters (Figure 1). Similarly, the passage of FD4 was not affected by IMQ (125 µM, either basolateral or apical; Figure 1). 

### 2.2. The Over-Stimulation of Colonic TLR7 with Imiquimod Does Not Trigger an Inflammatory-like Response within the Colon

Regardless of the experimental group, no signs of colonic inflammation were observed upon examination of the colon at the time of necropsy. Similarly, no change in colonic relative weight or the colonic expression of inflammatory markers (IL-6 and IFNα1) was observed among groups. Likewise, regardless of the treatments applied, no changes in TLR7 expression were detected (relative expression: vehicle, 1.07 ± 0.22; imiquimod 100 µg: 1.73 ± 0.48; imiquimod 300 µg, 0.78 ± 0.09; *p* = 0.601, ANOVA).

### 2.3. Pretreatment with Imiquimod Altered the Colonic Epithelial Conductance and Reduced Paracellular Permeability In Vitro 

In colonic samples obtained from animals exposed to imiquimod (300 µg/rat, intracolonic) the tissue conductance was slightly decreased, although statistical significance was not reached (*p* = 0.087 vs. vehicle-treated tissues). A lower dose (100 µg/rat, intracolonic) was without effect. Other epithelial electrical parameters were not affected by the pre-treatment with imiquimod (Figure 2A–C).

In imiquimod pre-exposed tissues, the apical-to-basolateral flux of FD4 was reduced in a dose-related manner. At the end of the experimental time (1 h) the passage of FD4 was reduced by 28.5% (*p* > 0.05) and 62.2% (*p* < 0.05) for the doses of 100 µg and 300 µg, respectively, when compared with vehicle pre-exposed tissues (Figure 1D).

In general, an increase in the short circuit current (I_sc_) in response to a single concentration of carbachol (CCh) (100 µM), added at the end of the experimental protocol, was observed regardless of the treatment applied. Occasionally (less than 10% of the preparations) the tissues were discarded because of the lack of response to CCh, as indicative of tissue damage.

### 2.4. Imiquimod Enhances Colonic Permeability after Disruption of the Epithelial Barrier in In Vivo Conditions 

In basal conditions (animals treated intracolonically with the vehicle), the passage of FD4 to blood and urine, as well as the accumulation in the colonic wall, was detected in a reproducible manner. Pre-treatment with intracolonic imiquimod (300 µg/rat) did not affect the passage of FD4 (Figure 3).

Intracolonic imiquimod followed by a challenge of the colonic mucosa with DMSO resulted in an enhanced accumulation of FD4 in the colonic wall, with a 2-fold increase in plasma levels (*p* < 0.05 vs. animals without DMSO) and an 8.5-fold increase in the FD4 concentration in urine (*p* < 0.05 vs. animals without DMSO) (Figure 2). DMSO, *per se*, resulted in a slight, non-significant, increase in the passage of FD4 to blood and urine (Figure 3). 

### 2.5. Effects of Imiquimod on Tight Junction-Related Proteins and Barrier-Modulators 

The expression of the main TJ-related proteins (occludin, tricellulin, claudin-2, claudin-3, JAM-1 and ZO-1) was detected in all colonic samples. Intracolonic imiquimod induced a dose-related down-regulation of ZO-1, achieving a 35.4% reduction at the 300 µg/rat dose (*p* < 0.05 vs. vehicle; Figure 4A). A similar trend was observed for the pore-forming protein claudin-2, but statistical significance was not achieved. Other TJ-related proteins were not affected by imiquimod (Figure 4A). Similar relative changes were observed when imiquimod was combined with the epithelial irritation with DMSO (Figure 4B), without consistent treatment-related changes in protein expression.

All the TJ-related proteins assessed were also detected using immunohistochemistry. Under the control conditions, distribution along the epithelium was protein-specific. Under control conditions, the claudin-2 expression was restricted to the lower-mid part of the colonic crypts (Figure 5). No treatment-associated changes were observed in the claudin-2 expression (intensity of staining) or distribution, without differences in the percentage of the crypt length with immunoreactivity. Claudin-3 immunoreactivity was distributed homogeneously in the colonic crypt, while forming intracellular granules located in the supranuclear cytoplasm in the superficial epithelial cells (Figure 5). Again, no treatment-associated changes in the distribution or intensity of staining were observed (Figure 5). ZO-1 was expressed along the whole crypt, forming a connecting net among epithelial cells; with no detectable changes associated with the treatments (Figure 5 and Figure 6). In all cases, immunoreactivity disappeared when the primary antibody was omitted, thus confirming the specificity of the staining (Figure 6).

The expression of proglucagon, precursor of the barrier-enhancer factor GLP-2, was detected in all colonic samples. No treatment-related changes were detected among groups; although a reduction in expression was observed for the highest dose tested, but without statistical significance (relative expression: vehicle, 1.25 ± 0.30; imiquimod 100 µg, 2.00 ± 0.69; imiquimod 300 µg, 0.82 ± 0.11; *n* = 5 per group; *p* = 0.082, ANOVA).

## 3. Discussion

In this study we show that the local, colonic, over-stimulation of TLR7 with the selective agonist imiquimod modulates the colonic EBF in rats. In vivo over-stimulation of TLR7 led to an enhancement of colonic epithelial barrier function, while acute, in vitro, stimulation was without effect. In order to verify these TLR7-dependent pro-barrier effects, over-stimulation of TLR7 was combined with the intracolonic administration of DMSO as a stimulus to alter epithelial permeability. Under these conditions, the opposite effect was observed, with an increase in epithelial permeability upon TLR7 stimulation. The effects of TLR7 on the EBF seem to be independent of the modulation of the expression or the distribution of the main tight-junction-related proteins or the modulation of the expression levels of the barrier-enhancer factor GLP-2.

Alterations of the colonic EBF are a component of several gastrointestinal diseases and are a common finding in IBD and IBS [1,2,3,4,5]. Although the pathophysiological mechanisms underlying these alterations are not completely understood, compelling evidence suggests the presence of a sustained enhancement of epithelial paracellular permeability facilitating the exposure to luminal antigens, triggering innate mucosal immune system responses, thus perpetuating a state of persistent, and abnormal, immune activation leading to the development of intestinal inflammation. Several studies suggest that gut-commensal microbiota and host–microbial interactions might be important pathogenic factors in this process [6,10,11,33,34,35,36,37,38,39]. Indeed, epithelial interactions with the microbiota and microbial-derived products, as well as the increased entrance within the intestinal wall, are altered in states of dysbiosis, which are likely contributing to the process of immune activation [3,24,40,41]. Nevertheless, the microbial contribution and the exact mechanisms involved are still unclear. In this context, in the present report, we explored the possibility that TLR7-dependent host–bacterial interactions might be part of the mechanisms through which the microbiota modulates epithelial barrier function. For this, we simulated a state of dysbiosis with over-stimulation of TLR7-dependent host–bacterial interactions through direct stimulation of TLR7 with a selective agonist, imiquimod, administered locally (intracolonically). We have previously used this approach to study TLR-dependent host–bacterial interactions in rats and mice [28,29]. 

The results obtained show that the local (colonic) over-stimulation of TLR7 with imiquimod leads to a reduction in the epithelial conductance and paracellular permeability to macromolecules in in vitro conditions (Ussing chamber), thus indicating an improvement of the EBF. These observations suggest that the over-stimulation of TLR7 associated with a dysbiotic state might have a defensive function, increasing the tightness of the epithelium and, therefore, preventing an increased passage of luminal, microbial-related antigens during dysbiosis. This protective role of TLR7 is in line with that observed in mice during DSS-induced colitis, where epithelial lesions, clinical signs and serum levels of pro-inflammatory markers were ameliorated after TLR7 stimulation with imiquimod [30]. A similar protective role has been postulated for other TLRs, mainly TLR2. Indeed, data obtained in murine models of dysbiosis indicate that TLR2 is a receptor that, upon activation by luminal microbial products, enhances the EBF as a protective response to dysbiosis, preventing the excessive passage of luminal antigens and an aberrant activation of the local immune system [24,42,43]. Altogether, these observations suggest that several TLRs, including at least TLR2 and TLR7, might share a similar protective role during states of over-stimulation (as potentially occurring during dysbiosis). Considering that that several TLRs have been largely associated with the activation of intestinal immune responses and the induction of inflammation and secretomotor alterations [23,44,45], these observations support a dual role for TLRs, mediating both pathogenic and protective responses in a TLR-specific manner. The balance between protective and damaging signals will lead to a final state in which homeostasis is maintained or functional alterations appear. 

Interestingly, the pro-barrier effects of TLR7 were not observed when the receptor was acutely stimulated in vitro (the direct addition of imiquimod in the Ussing chamber). This apparent discrepancy vs. the observation after the in vivo stimulation might suggest that the activation of TLR7 triggers a signaling cascade that requires a relatively long time-frame to generate functional changes. In this respect, the technical limitations of the technique used, since the viability of the mucosal sheets in the Ussing chamber is limited, might not allow the development of the full response observed after the in vivo stimulation. In addition, the recruitment of extra-intestinal neuroendocrine mechanisms, not preserved in in vitro conditions, might be necessary to elicit the full effects associated with TLR7 stimulation, which consequently, cannot be manifested during the acute in vitro stimulation.

Considering the potential protective role of TLR7, we hypothesized that its effects would manifest in states of barrier alteration in which the epithelial entrance of luminal factors might be facilitated. In these conditions, a more effective activation of TLR-dependent signaling mechanisms, including TLR7-mediated pro-barrier responses, might occur. To test this hypothesis, we assessed the effects of TLR7 over-stimulation during the simulation of a state of favored epithelial permeability, such as during DMSO exposure. In this sense, DMSO is an aprotic solvent that permeabilizes the cell membrane acting as a non-selective penetration enhancer [46]. In the presence of DMSO, we should expect an increased entrance of luminally-administered imiquimod and, therefore, an enhanced stimulation of intracellular TLR7 and the consequent activation of barrier protective mechanisms. However, against this hypothesis, the combined imiquimod—DMSO resulted in a deterioration of the EBF, with an increased passage of luminal macromolecules to blood and urine, as assessed in vivo. These effects are likely to be TLR7-dependent and not secondary to the permeabilization of the epithelium, since DMSO, *per se*, did not increase epithelial permeability to macromolecules. Currently, we cannot explain the differences observed between the stimulation of TLR7 by imiquimod alone or with the combination imiquimod–DMSO (a TLR7-mediated enhancement of the barrier vs. a TLR7-mediated worsening of the barrier, respectively). We can speculate that after DMSO administration the passage of luminal factors leads to the simultaneous activation of multiple mechanisms, including barrier protective and damaging pathways, with a final outcome depending upon the balance between both effects. 

In any case, the results obtained suggest that the TLR7-mediated effects on the EBF are mainly associated with a modulation of the permeability to macromolecules, which predominantly occurs through paracellular pathways. Since paracellular permeability to macromolecules depends largely on the TJ organized among the epithelial cells we also assessed if TLR7 over-stimulation could modify the expression of the main TJ-related proteins (occludin, tricellulin, claudin-2, claudin-3, JAM-1 and ZO-1). Overall, no consistent change in the expression of these proteins was observed following imiquimod administration, either alone or combined with DMSO. In spite of these negative findings, we cannot exclude conformational changes of the already available proteins that could lead to a reorganization of the TJ at a molecular level, thus explaining our functional findings. For instance, Cario et al. showed that the stimulation of TLR2 enhances the transepithelial resistance of the colonic epithelial barrier through the apical redistribution of ZO-1 [43]. Therefore, similar mechanisms cannot be discarded for TLR7. However, when assessing the epithelial distribution of ZO-1, claudin-2 and claudin-3 using immunohistochemistry, no consistent treatment-related change in the distribution or intensity of staining was observed during the stimulation TLR7 (with or without DMSO). Overall, these observations indicate that the TLR7-dependent modulation of the colonic EBF is independent of the changes in TJ-related proteins, at least as it relates to occludin, tricellulin, claudin-2, claudin-3, JAM-1 and ZO-1. Nevertheless, extended ultrastructural studies addressing the organization of TJ are necessary to further address the potential implication of TJ in the TLR7-mediated modulation of the EBF. 

GLP-2 has been postulated as a barrier-enhancer factor. Indeed, a positive correlation between intestinal proglucagon levels, the precursor of GLP-2, and the expression of the TJ-related proteins (ZO-1 and occludin) has been observed [31]. Similarly, a reduction in proglucagon expression has been described in states of increased epithelial permeability associated with the down-regulation of TJ-related proteins [12]. Despite this evidence, in our studies, no change in proglucagon gene expression was detected upon the activation of TLR7, regardless of the presence of changes in the EBF. This suggests that GLP-2 is not implicated in the barrier modulatory effect of TLR7.

In previous studies, we observed that intracolonic imiquimod had minor effects on immune activation in rats, with a moderate up-regulation of pro-inflammatory cytokines, observed only after repeated treatment [28,29]. In agreement with these observations, a single treatment with imiquimod resulted in neither the up-regulation of pro-inflammatory cytokines nor the induction of inflammatory like-changes within the colon, at either the macroscopical or microscopical levels. This indicates that the TLR7-mediated modulation of the EBF is not secondary to a local immune activation within the colon. These observations contrast with previous data showing that the TLR7/8 agonist resiquimod elicited a colitic-like state upon intracolonic administration in mice [47]. However, this effect is probably associated with the stimulation of TLR8 [48]. 

## 4. Materials and Methods

### 4.1. Chemicals

Imiquimod [R-837, 1-(2-Methylpropyl)-1H-imidazole [4,5-c]quinoline-4-amine]; Enzo Life Sciences, Farmingdale, NY, USA] was dissolved in 0.5% hydroxypropylmethyl cellulose (Sigma-Aldrich, Saint Louis, MO, USA). Carbachol (CCh, Sigma-Aldrich) and tetrodotoxin (TTX, Latoxan, Valence, France) were dissolved in distilled water as stock solutions of 10^−1^ M (CCh) and 10^−4^ M (TTX), respectively, and further dilutions were performed in distilled water. All stock solutions were aliquoted and stored at −80 °C until their use. Dimethyl sulfoxide (DMSO) was obtained from Panreac (Barcelona, Spain). Fluorescein isothiocyanate (FITC)-labeled dextran (FD) with a mean molecular weight of 4 kDa (FD4; TdB Consultancy AB. Uppsala, Sweden) was stored at 5 °C and was dissolved with Krebs solution (20 mg/mL) at the time of use.

### 4.2. Animals

Adult male Sprague–Dawley rats (6-week-old at arrival; 250–280 g; Charles-River Laboratories, Lyon, France) were used. A total of 109 animals were used for all the experiments described in this report, including the pilot studies and animals that were excluded because of technical problems (approximately 10%). On arrival, the animals were housed in pairs in standard plastic cages under conventional controlled environmental conditions (20–22 °C, 40–70% humidity and 12 h light/dark cycle) and fed with a standard pellet diet (Panlab SL, Barcelona, Spain) and tap water ad libitum. All experimental procedures were approved by the ethics committees of the Universitat Autònoma de Barcelona and the Generalitat de Catalunya (protocols 1420 and 6333, respectively).

### 4.3. Tissue Sampling

At the time of the experiments, animals were euthanized by decapitation, except when otherwise stated. A laparotomy was performed and the colon gently dissected and placed in ice-cold oxygenated Krebs buffer [(in mM) 115.48 NaCl, 21.90 NaHCO_3_; 4.61 KCl; 1.14 NaH_2_PO_4_; 2.50 CaCl_2_; 1.16 MgSO_4_ (pH: 7.3–7.4)] containing 10 mM glucose. Colonic segments were used to perform in vitro epithelial barrier function studies (mid colon) or preserved for morphological or molecular biology studies (mid-distal colon). Samples for histological and immunostaining studies were fixed in 4% paraformaldehyde in a phosphate buffer for 24 h. Thereafter, fixed samples were processed routinely for paraffin embedding and 5 µm sections were obtained for hematoxylin and eosin (H&E) staining or immunohistochemistry. Samples for molecular biology studies were immediately frozen in liquid nitrogen and stored at −80 °C until processed.

### 4.4. Measurement of Electrophysiological Parameters (Ussing Chambers)

Colonic segments were stripped of the outer muscle layers and myenteric plexus, opened along the mesenteric border, and divided into 1 cm^2^ flat segments, approximately. Epithelial sheets were mounted in Ussing chambers (World Precision Instruments, Aston, UK) with an exposed window surface area of 0.67 cm^2^. The tissues were bathed bilaterally with 5 mL of oxygenated and warmed (37 °C) Krebs buffer (in mmol/L: 115.48 NaCl, 21.90 NaHCO_3_, 4.61 KCl, 1.14 NaH_2_PO_4_, 2.50 CaCl_2_ and 1.16 MgSO_4_; pH: 7.3–7.4). The buffer also contained 10 mmol/L glucose as an energy source. The electrical equipment of the Ussing chambers consisted of two voltage-sensitive electrodes (EKV; World Precision Instruments) to monitor the potential difference (PD) across the tissue, and two Ag–AgCl current-passing electrodes (EKC; World Precision Instruments) to inject the required short-circuit current (I_sc_) to maintain a zero potential difference, as registered via an automated voltage/current clamp (DVC-1000; World Precision Instruments). Ohm’s law was used to calculate the tissue conductance (G), using the change in I_sc_ when a voltage step of 1 mV was applied at 5 min intervals. The tissues were allowed to stabilize for 15–25 min before the baseline values for PD, I_sc_ and G were recorded. The data were digitized with an analog-to-digital converter (MP150; Biopac Systems, Goleta, CA, USA) and measurements were recorded and analyzed with Acqknowledge computer software (version 3.8.1; Biopac Systems). The short circuit current and G were normalized for the mucosal surface area.

### 4.5. Epithelial Paracellular Permeability In Vitro

Paracellular permeability was evaluated following protocols previously described by us [12]. The mucosal-to-basolateral flux of fluorescein isothiocyanate (FITC)-labeled dextran (FD) with a mean molecular weight of 4 kDa was assessed in colonic epithelial sheets mounted in Ussing chambers. After stabilization (20–30 min), baseline electrophysiological parameters were assessed and FD4 were added to the mucosal reservoir to a final concentration of 2.5 × 10^−4^ M. Basolateral samples (250 µL, replaced by 250 µL of buffer solution with glucose) were taken at 15-min intervals during the following 60 min for the measurement of FD4. The concentration of fluorescein in the samples was determined by fluorometry (Infinite F200; Tecan, Crailsheim, Germany) with an excitation wavelength of 485 nm (20 nm band width) and an emission wavelength of 535 nm (25 nm band width), against a standard curve. Readings are expressed as percentage (%) of the total amount of FD4 added to the mucosal reservoir.

### 4.6. The In Vivo Colonic Permeability

To assess the colonic permeability in in vivo conditions, the passage of intracolonically administered FD4 to blood and urine, as well as the accumulation of FD4 in the colonic wall, was determined. For this, the animals were anesthetized with isoflurane (Isoflo^®^) and FD4 was administered intracolonically (rectal enema, 10 mg/animal, 0.2 mL) with a plastic cannula (8 cm from the anus, corresponding to the mid colon). FD4 administration was performed slowly (30 s–1 min) to avoid any reflux. Thereafter, the animals were returned to their home cages and 30 min later were deeply anesthetized with isoflurane and blood collected via intracardiac puncture. Subsequently, the animals were euthanized by a thoracotomy, and urine (intravesically) and colon samples were collected. The fluorescence was determined in urine, serum (obtained from blood centrifuged at 3000× *g* for 10 min, 4 °C) and colon homogenates as described above. When assessing the fluorescence in colonic samples, the tissues were washed thoroughly with a saline solution to eliminate any luminal remains of FD4.

### 4.7. Experimental Protocols

#### 4.7.1. The Effects of Imiquimod on Epithelial Electrical Parameters

The animals were deeply anesthetized with isoflurane (Isoflo^®^, Esteve Veterinaria, Barcelona, Spain) and treated with intracolonic imiquimod (rectal enema, 100 or 300 µg/rat) or vehicle (hydroxypropylmethyl cellulose; 5 mg/mL; 0.2 mL/rat). Five hours later, the animals were euthanized and colonic sheets were obtained and mounted in Ussing chambers as described above. After a 20 min stabilization period, the electrical parameters (I_sc_, PD and G) were assessed for an additional 60 min period.

In some cases, the effects of the acute exposure to imiquimod (as a direct addition to the Ussing chambers) on the electrical parameters were assessed in the colonic sheets obtained from a naïve animal. In this case, after a 20–30 min stabilization period the basal electrical parameters were recorded for a 10 min period, and thereafter, imiquimod (300 µg) was added to either the apical or the basolateral side and the changes in electrical parameters were recorded for an additional period of 60 min. The addition of the vehicle in some tissues served as a control.

At the end of the experiments, a single concentration of CCh (100 µM) was added to the basolateral side of the chamber in order to assess the viability of the tissues.

#### 4.7.2. The Effects of Imiquimod on Epithelial Permeability In Vitro

The isoflurane-anesthetized animals were treated with intracolonic imiquimod (100 or 300 µg/rat) or the vehicle (hydroxypropylmethyl cellulose; 5 mg/mL; 0.2 mL/rat). Five hours later, the animals were euthanized and the colonic sheets were obtained and mounted in Ussing chambers as described above. After a 20–30 min stabilization period, baseline electrophysiological parameters were assessed and FD4 was added to the mucosal reservoir (2.5 × 10^−4^ M) and the passage of the marker evaluated, as described above, for the following 60 min.

#### 4.7.3. The Effects of Imiquimod on Epithelial Permeability In Vivo

The rats were anesthetized with isoflurane and treated intracolonically with imiquimod (300 µg/rat) or its vehicle (0.2 mL) and returned to their home cages. Six hours later, the animals were anesthetized again and FD4 (10 mg/rat, 0.2 mL) was administered intracolonically and permeability tested 30 min later, as described above.

In some cases, the imiquimod treatment was combined with a permeabilization of the colonic epithelium. In these cases, 4 h after the imiquimod administration, animals were anesthetized with isoflurane and the colonic epithelium was challenged with dimethyl sulfoxide [46] (DMSO, 100%, 0.2 mL) and 2 h later FD4 was administered intracolonically to assess permeability.

In all cases, the animals received a single intracolonic treatment. For this, the animals were deeply anesthetized with isoflurane (Isoflo^®^, Esteve Veterinaria, Barcelona, Spain) and treatments were applied intracolonically with a plastic cannula (8 cm from the anus, corresponding to the mid-distal colon). Thereafter, the animals were returned to their home cages and 3 h (gene expression), 5 h (Ussing chamber studies) or 6.5 h later (in vivo permeability) were euthanized and tissue samples were obtained as described above.

### 4.8. The Quantitative Real Time Reverse Transcription Polymerase Chain Reaction (RT-qPCR)

The total RNA was extracted from colonic tissue samples using Ribopure RNA Isolation Kit (Applied Biosystems, CA, USA) and quantified with a Nanodrop (ND-100 spectrophotometer, Nanodrop Technologies, Rockland, DE, USA). For cDNA synthesis, 1 µg of RNA was reverse-transcribed in a 20 µLl reaction volume using a high capacity cDNA reverse transcription kit (Applied Biosystems, Foster City, CA, USA). TaqMan gene expression assays for occludin (Rn00580064_m1), tricellulin (Rn01494284_m1), claudin-2 (Rn02063575_s1), claudin-3 (Rn00581751_s1), JAM-1 (Rn00587389_m1), ZO-1 (Rn02116071_s1), IL-6 (Rn01410330_m1), IFNα (Rn02395770_g1), TLR 7 (Rn01771083_s1) and proglucagon (Rn00562293_m1) were used (all from Applied Biosystems). Actin-β (Rn00667869_m1) was used as an endogenous reference gene. The PCR reaction mixture was incubated on a 7500 Fast Real Time PCR system (Applied Biosystems). All samples, as well as the negative controls, were assayed in triplicates. The cycle threshold for each sample was obtained and, thereafter, all data were analyzed with the comparative 2^−∆∆CT^ method, with the control group serving as the calibrator [49].

### 4.9. Immunohistochemistry for Tight-Junction-Related Proteins

Paraffin embedded tissue sections (5 µm thick) were deparaffinized and rehydrated with a battery gradient of alcohols. The antigen retrieval for claudin-2 and claudin-3 was achieved by processing the slides in a microwave (2 cycles of 5 min, 800 W) in a 10 mM Tris Base, 1 mM EDTA solution (pH 9). The epitope retrieval for ZO-1 was performed using a pressure cooker (at full pressure for 6 min) in 10 mM citrate buffer (pH 6). Thereafter, samples were incubated for 40 min in H_2_O_2_ (5% in distilled water) for inhibiting endogenous peroxidases and with the reagents of the Avidin/Biotin Blocking Kit (SP-2001; Vector Laboratories, Burlingame, CA, USA) for inhibiting the endogenous avidin and biotin. Finally, a 1 h incubation at room temperature with horse, rabbit or goat serum, as appropriate, was performed for blocking unspecific unions before incubating the slides (overnight at 4 °C) with their respective primary antibodies: mouse monoclonal anti-Claudin-2 antibody (1:2000; ref.: 32–5600. Invitrogen, Camarillo, CA, USA), goat polyclonal anti-Claudin-3 antibody (1:500; ref.: SC-17662. Santa Cruz, Dallas, TX, USA), rabbit polyclonal anti-Occludin (1:1000; ref.: SC-5562. Santa Cruz) and rabbit polyclonal anti-ZO-1 (1:500; ref.: SC-10804. Santa Cruz). The following day, sections were incubated with the respective secondary antibody for 1 h at room temperature: biotinylated horse anti-mouse IgG (1:200, ref.: BA-2000, Vector Laboratories), biotinylated rabbit anti-goat IgG (1:200, ref.: SC2774. Santa Cruz) or Biotin-XX Goat Anti-Rabbit IgG (H + L) (1:200, ref.: B2770. Invitrogen). In all cases, an avidin/peroxidase kit (Vectastain ABC kit; Vector Laboratories, Burlingame, CA, USA) was used for detection, and the antigen–antibody complexes were revealed using 3,3′-diaminobenzi-dine (SK-4100 DAB; Vector Laboratories). The slides were counterstained with toluidine blue or with hematoxylin. In all cases, the specificity of the staining was confirmed by omitting the primary antibody.

For the claudin-2 and ZO-1 immunohistochemistry results, 10–20 representative microphotographs (×200) were taken per animal with a Nikon Eclipse 90i microscope. The intensity of the staining was measured using the software Image J (National Institute of Health, Bethesda, MD, USA) and a mean value obtained. Claudin-3 immunoreactivity at the epithelial surface was quantified by applying a semi-quantitative score (0: no immunoreactivity; 1: scarce, low intensity, granules in some epithelial cells; 2: clear, but not-organized granules in abundant epithelial cells; 3: clear and well-organized granules, drawing a chain in the zone corresponding to the epithelial TJs, in most cells). In this case, for quantification, 15 randomly selected fields covering the whole thickness of the jejunal mucosa, from at least two tissue sections for each animal, were scored by two independent observers. A final score (0–3) for each animal was calculated as the mean of the scores assigned by each observer. All procedures were performed on coded slides to avoid any bias.

### 4.10. Statistical Analysis

All data are expressed as a mean ± SEM. A robust analysis (one interaction) was used to obtain the mean ± SEM for the RT-qPCR data. Comparisons between two groups were performed using Student’s unpaired *t* test. Comparisons between multiple groups were performed using a one-way or a two-way ANOVA, as appropriate; followed, when necessary, by a Newman–Keuls’ multiple comparisons test. In all cases, the results were considered statistically significant when *p* < 0.05. A GraphPad Prism 4 (GraphPad Software, La Jolla, CA, USA) was used to perform all statistical analyses.

## 5. Conclusions

Overall, we show that the colonic activation of TLR7 might be associated with an enhancement of the EBF in normal conditions. On the other hand, a deterioration of the barrier function was observed in states of epithelial permeabilization, which are potentially associated with an increased stimulation of the receptor and/or a negative balance between the protective–detrimental factors in states of epithelial dysfunction. The mechanisms underlying these effects remain unclear. The results obtained suggest that neither the modulation of the expression and/or organization of the main tight junction-related proteins, nor the changes in the expression of proglucagon, as a precursor of the barrier-enhancer GLP-2, mediate the observed changes in the EBF. These observations, together with previous data, suggest that the TLRs-mediated host–bacterial interactions might elicit protective or detrimental responses from the epithelial barrier function. The modulatory effects of TLRs on the EBF might contribute to the pathophysiological alterations observed in gastrointestinal diseases, which have dysbiosis and barrier alterations as common components, such as IBD or IBS.

## Figures and Tables

**Figure 1 ijms-24-01254-f001:**
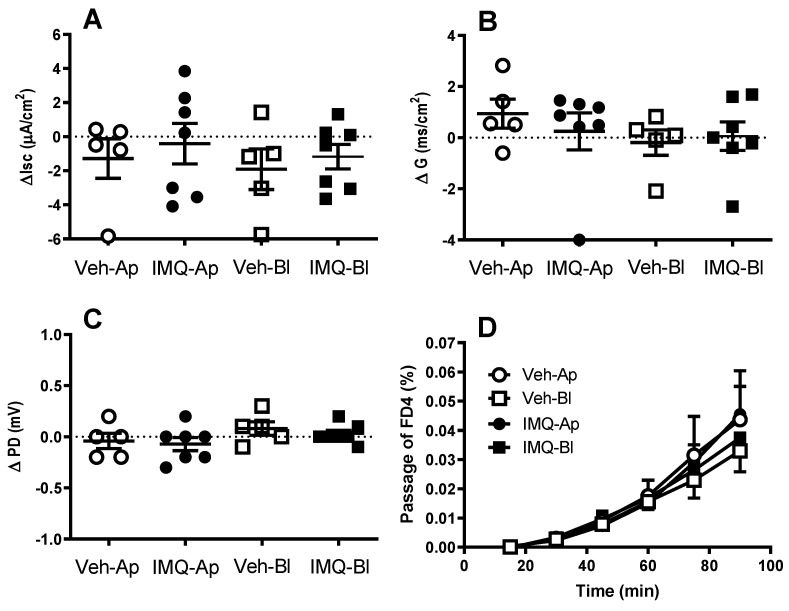
The effects of the acute addition of imiquimod (125 µM, IMQ) or its vehicle (0.5% hydroxypropylmethyl cellulose, Veh) to the Ussing chamber, either to the apical (Ap) or the basolateral (Bl) side, on the colonic epithelial electrical parameters and permeability to macromolecules. (**A**–**C**) the net change in epithelial electrical parameters: short circuit current (I_sc_, panel **A**), conductance (G, panel **B**), potential difference (PD, panel **C**). Each point represents a colonic sheet, and the horizontal line with errors shows the mean ± SEM (*n* = 5–7 colonic sheets from 5 animals per group). The net change (∆) was obtained as the difference between the value before and the value after the 60 min exposure to imiquimod. (**D**) Effects of imiquimod on epithelial permeability to macromolecules. The graph shows the mucosal to basolateral passage (as a percentage of the amount added to the mucosal reservoir) of fluorescein isothiocyanate-dextran 4 kD (FD4) in tissues exposed to Sthe vehicle (Veh) or imiquimod (125 µM, IMQ), either to the apical (Ap) or the basolateral (Bl) side. Data are mean ± SEM, *n* = 4–7 colonic sheets from 3–6 animals per group.

**Figure 2 ijms-24-01254-f002:**
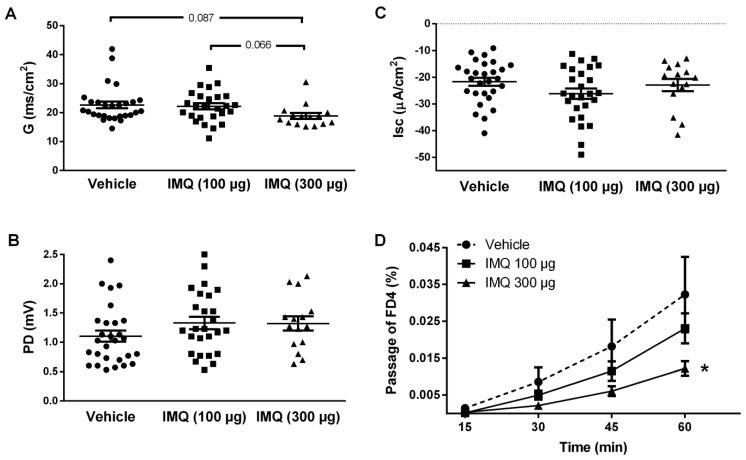
Effects of pre-exposure to imiquimod (IMQ) (intracolonic) on the colonic epithelial electrical parameters and permeability to macromolecules. (**A**–**C**) Epithelial electrical parameters: conductance (G, panel **A**), potential difference (PD, panel **B**) and short circuit current (I_sc_, panel **C**). Each point represents a colonic sheet, and the horizontal line with errors shows the mean ± SEM (*n* = 15–28 colonic sheets from 9–16 animals per group). (**D**) Effects of imiquimod on epithelial permeability to macromolecules. The graph shows the mucosal to basolateral passage (as percentage of the amount added to the mucosal reservoir) of fluorescein isothiocyanate-dextran 4 kD (FD4) in control conditions (vehicle pre-treatment) and in tissues pre-exposed to imiquimod. Data are the mean ± SEM, *n* = 6–12 colonic sheets from 3–6 animals per group. *: *p* < 0.05 vs. Vehicle.

**Figure 3 ijms-24-01254-f003:**
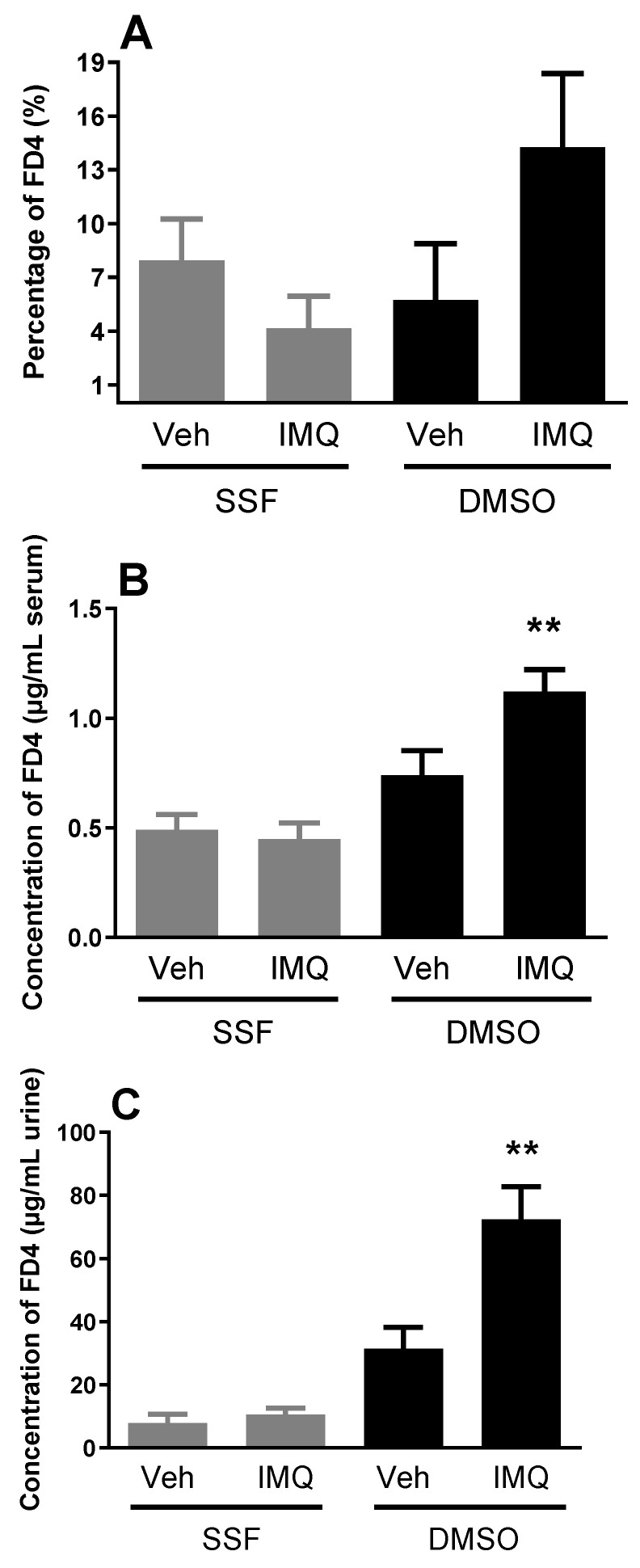
Effects of imiquimod (IMQ) on colonic permeability to FD4 in in vivo conditions. Data show the accumulation of FD4 on the colonic wall (**A**) and the passage of FD4 from colon to blood (µg of FD4/mL serum, (**B**)) and urine (µg of FD4/mL urine, (**C**)) during a 30-min period in control conditions and after treatment with imiquimod with or without DMSO. Data are mean ± SEM, *n* = 9–11 animals per group. **: *p* < 0.01 vs. other groups (ANOVA). Veh: Vehicle. SSF: Saline. DMSO: Dimethyl sulfoxide.

**Figure 4 ijms-24-01254-f004:**
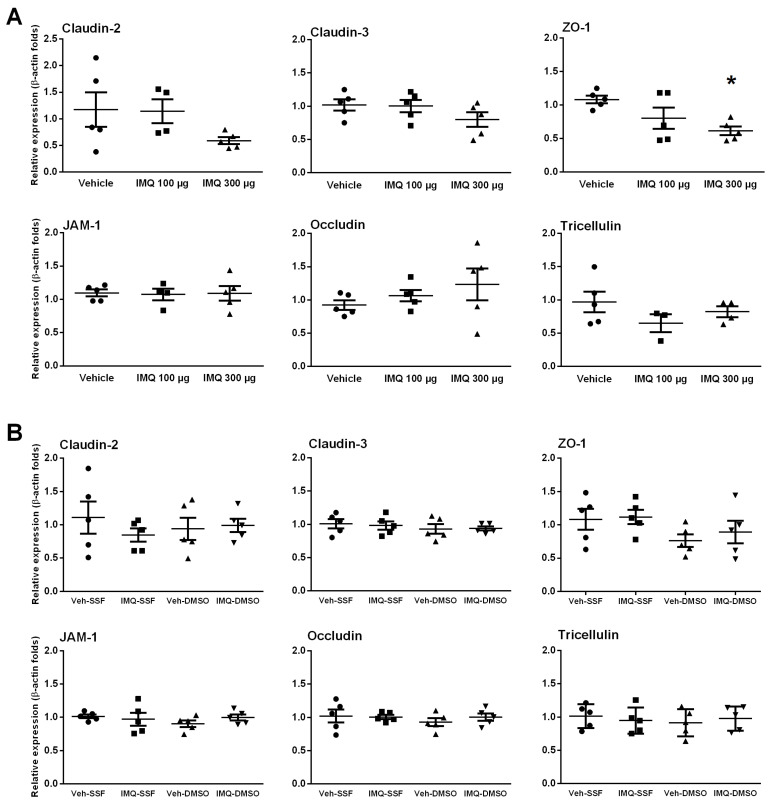
The effects of imiquimod (IMQ) on the gene expression of colonic tight junction-related proteins (occludin, tricellulin, claudin-2, claudin-3, JAM-1 and ZO-1). (**A**) Gene expression in tissues at 3 h post-intracolonic treatment with imiquimod. (**B**) Gene expression in animals treated with imiquimod combined with an epithelial challenge with DMSO. Each symbol represents an individual animal (*n* = 4–5 per group), the horizontal lines with errors correspond to the robust mean ± SEM. *: *p* < 0.05 vs. vehicle (ANOVA).

**Figure 5 ijms-24-01254-f005:**
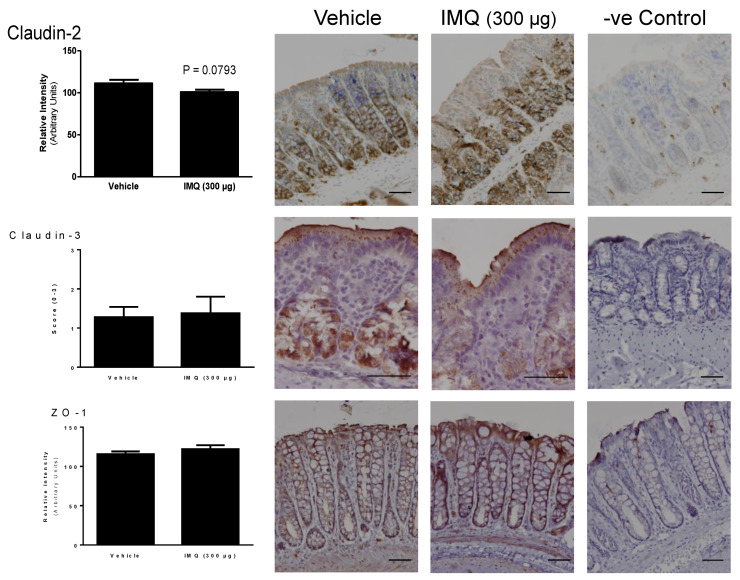
Immunohistochemistry for tight junction-related proteins: claudin-2, claudin-3 and ZO-1. Left column: quantification of immunostaining in control and imiquimod (IMQ)-treated rats (300 µg/rat, intracolonic). See the Materials and Methods section for details of quantification. Data are the mean ± SEM, *n* = 8–13 per group. Microphotographs: Representative images of a control animal (vehicle), an imiquimod (300 µg, intracolonic)-treated animal and a negative control (-ve control) for ZO-1, claudin-2 and claudin-3. Scale bar: 50 µm.

**Figure 6 ijms-24-01254-f006:**
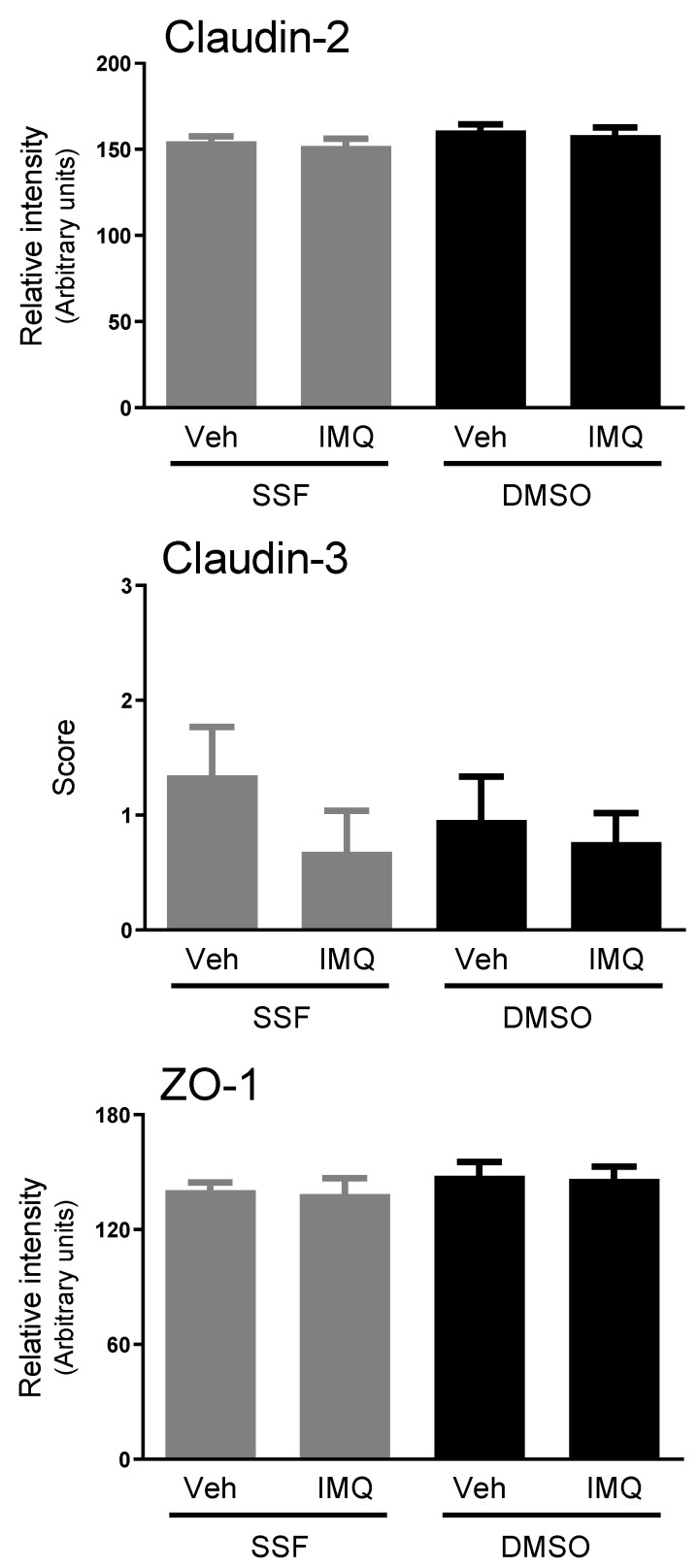
Quantification of immunostaining for tight junction-related proteins (claudin-2, claudin-3 and ZO-1) in control and imiquimod-treated rats with or without DMSO. See the Materials and Methods section for details of quantification. Data are the mean ± SEM, *n* = 3–5 per group. Veh: Vehicle. SSF: Saline. DMSO: Dimethyl sulfoxide.

## Data Availability

All data presented in this study are available on request from the corresponding author (vicente.martinez@uab.es).

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
