# Peer review of "The Local Activation of Toll-like Receptor 7 (TLR7) Modulates Colonic Epithelial Barrier Function in Rats"

_ijms, 2023, doi:10.3390/ijms24021254_

Round 1
Reviewer 1 Report
The work reported by Estevez and Martinez is an important addition to the role of TLR7 on colonic epithelial barrier functions. I recommend publishing after addressing the points below.
1. The chemical used in the study were purchased from vendors as listed in the experimental. After preparing solutions, how were these solutions stored?
2. If DMSO and SSF subgroups in fig. 2 can be presented in different colors, this will help with clarity.
3. The abstract can be more concise
Reviewer 2 Report
Review Report
Manuscript ID: ijms-2120467
Title: Local Activation of Toll-like Receptor 7 (TLR7) Modulates Colonic Epithelial Barrier Function in Rats
The aim of presented study was to investigate the effects of stimulation of Toll-like Receptor 7 on colonic epithelial barrier function in rats. The authors performed in vitro and in vivo experiments and presented results are interesting, however some issues need to be clarify before making decision on publication the manuscript.
My decision: major revision
The Authors should follow the comments and suggestions that are given below:
Abstract: The Authors’ affiliation should be in English;
Results:
Section 2.2.: (1) Lines; 80-84: I suggest to slightly extend the description of the electrical parameters. Despite the lack of significant changes, it would be valuable to present i.e. the mean values of results obtained. (2) I strongly suggest to add the information on the results of IL-6, INFα1 and TLR7 gene expression to the main text. (3) The legends on the Y-axis at Figure 2B and Figure 2C should be supplemented with “serum” and “urine”, respectively; (3) The charts and pictures presenting the results of TJ-related proteins (Figure 4) should be re-organized according to the description in the text. (4) Why different methods of quantification (relative intensity and scoring) were used for immunostaining of ZO-1 and Claudin-2 versus Claudin-3? (5) I suggest to add to the main text the figure presenting the results of proglucagon gene expression; (6) The results of ZO-1, Claudin-3 and Claudin-3 immunostaining in control and IMQ-treated rats with or without DMSO (presenting at Figure 5) are based on different number of rats (from 3 to 5 animals per group). In my opinion the results based only on 3 animals can be unreliable. What was the reason for the different numbers of animals in the groups?
“Conclusions” should be a separate section. The authors should clearly indicate the scientific novelty and originality of the presented results.
Materials and methods:
- (1) The graphical scheme presenting the experiments (including the numbers of animals used in each Exp.) is strongly recommended; (2) the subsection describing the chemicals and reagents should be at the beginning of this section; (3) Referring the ethical issues, the name of ethical committee provided in the text is very general, therefore please confirm that the Committee given in the manuscript deals strictly with the evaluation and approvement of animal experiment (or please provide the exact name of this authority); (4) The body weights of animals were not indicated in the text (5) I did not find any information about the number of rats used in the experiment (Total and in experimental groups).(6) The rats were exposed to IMQ at a dose of 100 or 300 µg/animal. Why the does of IMQ was not relative to animals weight? (7) How was the Imiquimod prepared for the animals experiment? How was a volume of IMQ administered to rats?
Round 2
Reviewer 2 Report
The Authors addressed all my comments and suggestions. I recommend the text for publication.